# High Diversity of Human Non-Polio Enterovirus Serotypes Identified in Contaminated Water in Nigeria

**DOI:** 10.3390/v13020249

**Published:** 2021-02-05

**Authors:** Manasi Majumdar, Dimitra Klapsa, Thomas Wilton, Erika Bujaki, Maria Dolores Fernandez-Garcia, Temitope Oluwasegun Cephas Faleye, Adefunke Olufunmilayo Oyero, Moses Olubusuyi Adewumi, Kader Ndiaye, Johnson Adekunle Adeniji, Javier Martin

**Affiliations:** 1Division of Virology, National Institute for Biological Standards and Control (NIBSC), Potters Bar EN6 3QG, Hertfordshire, UK; manasi.majumdar@nibsc.org (M.M.); dimitra.klapsa@nibsc.org (D.K.); thomas.wilton@nibsc.org (T.W.); erika.bujaki@nibsc.org (E.B.); 2Department of Virology, Institute Pasteur, Dakar, Senegal; dolores.fernandez@yahoo.es (M.D.F.-G.); kader.ndiaye@pasteur.sn (K.N.); 3Department of Virology, Faculty of Basic Medical Sciences, College of Medicine, University of Ibadan, Ibadan, Oyo State, Nigeria; faleyetemitope@gmail.com (T.O.C.F.); adewumi1@hotmail.com (M.O.A.); adek1808@yahoo.com (J.A.A.); 4World Health Organization National Polio Laboratory, Ibadan, Oyo State, Nigeria; funkeao@yahoo.com

**Keywords:** environmental surveillance, wastewater, human enterovirus, next generation sequencing (NGS), sewage, recombination, whole-genome sequencing

## Abstract

Human enteroviruses (EVs) are highly prevalent in sewage and have been associated with human diseases with complications leading to severe neurological syndromes. We have used a recently developed molecular method to investigate the presence of EVs in eight samples collected in 2017–2018 from water streams contaminated by drainage channels in three different locations in Nigeria. A total of 93 human EV strains belonging to 45 different serotypes were identified, far exceeding the number of strains and serotypes found in similar samples in previous studies. Next generation sequencing analysis retrieved whole-capsid genomic nucleotide sequences of EV strains belonging to all four A, B, C, and D species. Our results further demonstrate the value of environmental surveillance for the detection of EV transmission of both serotypes commonly associated with clinical syndromes, such as EV-A71, and those that appear to circulate silently but could eventually cause outbreaks and disease. Several uncommon serotypes, rarely reported elsewhere, were detected such as EV-A119, EV-B87, EV-C116, and EV-D111. Ten EV serotypes were detected in Nigeria for the first time and two of them, CV-A12 and EV-B86, firstly described in Africa. This method can be expanded to generate whole-genome EV sequences as we show here for one EV-D111 strain. Our data revealed phylogenetic relationships of Nigerian sewage strains with EV strains reported elsewhere, mostly from African origin, and provided new insights into the whole-genome structure of emerging serotype EV-D111 and recombination events among EV-D serotypes.

## 1. Introduction

Water is an essential resource used for critical human activities such as drinking, cooking, washing, recreational events, crop irrigations, and industrial use. While developed countries have made significant progress in providing adequate quality water and sanitation to most households, many of the developing countries are still deficient in these areas, leading to many deaths related to the ingestion of virus-contaminated water and other waterborne pathogens. It is understood that an estimated 663 million people globally regularly consume untreated water obtained from different sources including groundwater and surface water [1]. Contaminated water remains a leading source of diseases caused by viruses and bacteria such as polio, typhoid fever, diarrhea, and dysentery in many parts of the world. Among them, human enteroviruses (EVs) are highly prevalent in sewage and have been associated with human diseases with complications leading to severe neurological syndromes.

EVs are common pathogens that have co-evolved with humans for many years transmitting mainly through the fecal-oral or respiratory routes with use of poor-quality water supply and sanitation being associated with high virus transmission. The EV genome consist of a single RNA positive strand molecule approximately 7400 nucleotides in length that contains a region coding for structural and non-structural proteins flanked by 5′ and 3′ short non-coding regions [2]. A new alternative short open reading frame coding for a short 56–76 amino acid peptide possibly involved in cell tropism has been recently described [3]. There are at least 110 different EV serotypes grouped in four different species, A to D [4]. EV strains from all four species have been associated with a variety of disease syndromes including meningitis, myocarditis, and severe respiratory disease [5]. Complications sometimes lead to neurological syndromes, the most notorious being acute flaccid paralysis (AFP) caused by poliovirus, which is the subject of an intense global eradication campaign led by WHO that has brought the virus to the brink of extinction [6]. Apart from poliovirus for which standardized surveillance protocols and quality performance indicators have been defined, there is no systematic surveillance for any other EV serotype despite many examples of some serotypes causing severe disease, including neurological complications, such as EV-A71 and EV-D68 causing outbreaks associated with polio-like paralytic cases in recent years [7,8]. For this reason, information on EV serotypes infecting humans, although relatively abundant, is patchy and does only provide a limited view of the actual EV circulation patterns in humans. Indeed, only one or few sequences of EV strains are available for many of the 110 EV serotypes that have been described making it difficult to understand their prevalence and possible implication in human disease. 

As an alternative to clinical diagnosis, environmental surveillance (ES) has revealed as a very sensitive method for EV detection, particularly for poliovirus, which can be readily identified in sewage and water systems contaminated with sewage even in the absence of paralytic disease [9,10,11,12]. There is plenty of evidence of ES being able to track down the interruption of poliovirus transmission detecting both vaccine viruses following immunization campaigns or vaccine cessation and wild-type or vaccine-derived polioviruses during outbreaks [13]. The use of molecular methods in combination with next generation sequencing (NGS) analysis directly from clinical and environmental samples, without the use of cell culture virus isolation, by us and others is starting to unravel the complexity of the human virome in wastewater samples [14,15,16,17,18,19]. 

With a view to investigate the prevalence and diversity of EVs in samples from water streams often contaminated by drainage channels in Nigeria, we have used a recently developed NGS method to sequence PCR products synthesized directly from water concentrates retrieving whole-capsid sequences of EV strains from all four A to D species [16,17]. EV nucleotide sequences determined from water samples were used to identify EV serotypes and to establish phylogenetic relationships with EVs described elsewhere. The whole-genome sequence of a virus strain from emerging serotype EV-D111, identified in one of the samples, was also determined using overlapping PCR products. This sequence was compared with sequences of EV strains from all species D serotypes including recent EV-D111 isolates [20] in order to explore the occurrence of recombination events among EV-D serotypes.

## 2. Materials and Methods

### 2.1. Water Sample Collection and Processing

One liter grab water samples were collected at random from water bodies such as streams and rivers, presumably contaminated by drainage channels, at three different locations in Nigeria during 2017 and 2018 (Table 1). Community dwellers often interact with these water bodies and conceivably use for domestic purposes. Raw water samples were concentrated using a two-phase aqueous separation system following WHO recommended protocols [9]. A volume of around 10 mL was obtained from the 500 mL of crude starting material, thus attaining a concentration factor of 50×. Water concentrates that had been found to contain non-polio EVs following the recommended WHO testing algorithm used for poliovirus surveillance [21,22], that includes cell culture and molecular typing assays were selected for our analysis. 

Samples that did not contain poliovirus were specifically selected in order to focus on non-polio EV serotypes and avoid biosafety containment issues. Although, it cannot be completely ruled out, we do not expect bias in EV serotype distribution due to selecting poliovirus-negative samples in our study. First, the workflow used for all samples was identical, following WHO Guidelines for environmental surveillance [9]. Second, both polio and non-polio EVs have similar structural and physicochemical properties and are expected to be affected by dilution and inactivation effects in sewage contaminated water in a similar manner. Hence, identification of any EV is used and has proven to be a good indicator of good environmental sensitivity for poliovirus detection regardless of EV serotypes present in the sample [23]. Lastly, recent studies suggest there is little interference between the circulation of different EV serotypes in the same population with circulation of specific EV serotypes not having a major impact on the transmission of other EV serotypes [24]. Indeed, several other factors do seem to have an effect on the detection of EVs in environmental samples such as the nature of the sewage network, the appropriateness of the sampling site, the catchment population, and the physicochemical properties of sewage [25]. Samples used in this study were taken from water streams that are often contaminated with sewage, but levels of contamination and catchment population numbers are difficult to estimate making that comparisons between sites more difficult. Water concentrate samples were shipped from the Polio Laboratory in Nigeria to the National Institute for Biological Standards and Control (NIBSC) in dry ice and were stored at −80 °C until use. Whole-capsid EV sequences present in water concentrates were determined by NGS analysis of whole-capsid PCR products using methods described in Section 2.2, Section 2.3 and Section 2.4. These methods have been extensively validated using reference EV strains, laboratory-made EV mixtures, clinical samples, and water concentrates from different locations in comparison to the sequences determined by the Sanger method [16,17,19].

### 2.2. Pan-EV Whole Capsid-coding Region RT-PCR Amplification

We followed a method designed to amplify the entire capsid sequences of EV strains from all four A, B, C, and D described before [16], which involves two independent PCR reactions. The products from the two PCR reactions were mixed and analyzed in a single NGS reaction. Viral RNA was extracted from the water concentrate using Roche High Pure viral RNA kit (Roche Life Science, Mannheim, Germany). RT-PCR fragments covering the entire capsid-coding region and part of the region coding for non-structural proteins (2A–2C) were synthesized from purified viral RNAs by one-step RT-PCR using a SuperScript III One-Step RT-PCR System with Platinum Taq High Fidelity DNA Polymerase (Thermo Fisher Scientific, Waltham, MA, USA). Two different reactions were performed using two primer sets:Primers 5′NCR (5′-TGGCGGAACCGACTACTTTGGGTG-3′) and CRE-R (5′-TCAATACGGTGTTTGCTCTTGAACTG-3′)Primers MM_EV_F2 (5′-CAGCGGAACCGACTACTTT-3′) and MM_EV_R1 (5′-AATACGGCATTTGGACTTGAACTGT-3′)

Reaction conditions were: 50 °C for 30 min followed by 94 °C for 2 min plus 42 cycles of 94 °C for 15 s, 55 °C for 30 s, and 68 °C for 8 min with a final extension step of 68 °C for 5 min. Amplified products from both reactions were purified using AMPure XP magnetic beads (Beckman Coulter, Indianapolis, IN, USA) and pooled (1:1) before being analyzed by NGS. The expected amplicon size for both RT-PCR reactions is approximately 3950 nucleotides (nucleotides 553–4459, numbering as in PV1 Sabin AY184219 reference strain).

### 2.3. Generation of Sequencing Libraries and Quality Trimming of NGS Reads 

Sequencing libraries were prepared using Nextera XT reagents and sequenced on a MiSeq using a 2 × 301-mer paired-end v3 Flow Cell and the manufacturer’s protocol (both Illumina, San Diego, CA, USA). NGS sequencing data were processed using Geneious 10.2.3 software package (Biomatters, Auckland, New Zealand) as described before [16]. Briefly, raw sequence data were imported into Geneious 10.2.3 and paired end reads combined. Data were filtered using a custom workflow with the following parameters: PCR primers and Nextera adaptor/index sequences were trimmed from 5′ and 3′ ends with a minimum 5 bp overlap; reads were trimmed to have no bases with a quality <Q30 and no ambiguities. Following this, reads <50 nt in length were discarded and duplicate reads were removed using the program Dedup (within Geneious 10.2.3). 

### 2.4. Generation of Whole-Capsid EV Sequence Contigs by de Novo Assembly of Filtered NGS Reads

The filtered NGS reads were then assembled de novo using stringent assembly conditions as described before [26]: minimum 50 base overlap, minimum overlap identity of 98%, maximum 2% mismatches per read, and only using paired hits during assembly. In addition, the options to produce scaffolds and ignore words repeated more than 100–1000 times, available in the Geneious 10.2.3 assembler, were selected to improve the quality of assembly. Following BLAST analysis, only >2500 nt per site covering the capsid coding region were selected for further analysis. Filtered reads were finally mapped to these selected contigs to obtain final consensus sequences by assigning the most common nucleotide sequence to each nucleotide position. Manual analyses for visualizing and quantifying assembly results were performed throughout the process. As a result, we determined nucleotide sequences of a number of EV strains in each sample. This method is equivalent to operational taxonomic unit (OTU) clustering that has been described to identify and classify bacteria and viruses in biological samples, with sequences being clustered according to their similarity to one another, and OTUs defined based on a high similarity threshold (usually between 97% and 100% similarity). Our method can discriminate between genotypes and sub-genotypes within serotypes and has been extensively validated using laboratory mixtures of reference EV strains of known sequence and comparison with Sanger sequence analysis and analysis using the Oxford Nanopore MinION system which allows sequencing complete PCR products. 

### 2.5. Phylogenetic Analysis of EV Strains 

EV sequences obtained in this study were compared to those of other EVs available in the NCBI database. The closest virus relatives to the Nigerian EV environmental strains were identified using the RIVM and BLAST online sequence analysis tools [27,28] and EV serotypes were assigned on the basis of their VP1 sequence. EV genome sequences were aligned using the program ClustalW (within Geneious 10.2.3). Molecular Evolutionary Genetics Analysis (MEGA) software package version X [29] was used for phylogenetic analyses. Phylogenetic relationships between EV sequences were inferred by using the Maximum Likelihood method and Tamura-Nei model using MEGA X software. Sequence divergence was determined by calculating mean pairwise distances. Initial tree(s) for the heuristic search were obtained automatically by applying Neighbor-Join and BioNJ algorithms to a matrix of pairwise distances estimated using the Maximum Composite Likelihood (MCL) approach, and then selecting the topology with superior log likelihood value. Phylogenetic trees were drawn using FigTree version 1.4.0 program (http://tree.bio.ed.ac.uk/software/figtree/).

### 2.6. Whole-Genome Amplification and Sequencing of EV-D111 Strain from KAT-17-1263 Water Sample

The nearly complete genome of EV-D111 strain from KAT-17-1263 sample was determined by NGS analysis of overlapping PCR products. Primers D111-WGF (5′-CAGGAAACAGCTATGACCTTAAAACAGCYYKDGGGTTG-3′) and D111-IR (5′-GCACATAAACGGTATGGTCATTCTAGCTGG-3′) were used to generate a PCR product covering the 5′-end half of the genome and primers D111-IF (5′-CCAGCTAGAATGACCATACCGTTTATGTGC-3′) and D111-WGR (5′-TTCCCAAKTRACCAAAATTTACCTC-3′) were used to generate a PCR product covering the 3′-end of the genome. Amplification conditions and NGS analysis to obtain consensus sequences for each RT-PCR product were the same as those described in Section 2.2, Section 2.3 and Section 2.4. Finally, the two overlapping contigs, together with the PanEV whole-capsid contig containing reads mapping to the EV-D111 sequences were assembled to produce the whole-genome sequence. 

### 2.7. Whole-Genome Amplification and Sequencing of EV-D94 Isolate from the Republic of Niger

The nearly complete genome of EV-D94 was determined in a different project characterizing EVs present in infected cell cultures from clinical stool samples from AFP surveillance. As it is expected that such cell culture samples contain high concentration of virus, we opted for sequencing PCR products generated by random primers as opposed to specific primers required to amplify low concentrations of viral RNA present in environmental samples. PCR fragments for sequencing were generated by sequence-independent single-primer amplification (SISPA) of purified RNAs from infected cells and NGS analysis was performed as described before [30,31]. RT-PCR templates were generated in a random RT-PCR reaction using primers RA01-N8 (5’-GCC GGA GCT CTG CAG ATA TCN NNN NNN N-3′) and RA01 (5´-GCC GGA GCT CTG CAG ATA TC-3´). Sequencing libraries were prepared using Nextera XT reagents and sequenced on a MiSeq using a 2 × 301-mer paired-end v3 Flow Cell and manufacturer’s protocols (both Illumina, San Diego, CA, USA). Quality trimming of NGS reads was performed as described in Section 2.3. Host contaminating human genome sequences were filtered out using the hg38 database 27. Filtered reads were then assembled de novo using stringent assembly conditions as described in Section 2.4 and the resulting contigs were analyzed by BLAST analysis. 

### 2.8. Recombination Analysis

The whole-genome sequences of EV-D11-KAT-17-1263 and EV-D94-14-534 strains were aligned to those from representatives of Species D EV serotypes D68, D70, D94, and D111 using the program ClustalW (within Geneious 10.2.3) and subjected to recombination analysis using Simplot version 3.5.1 software (https://sray.med.som.jhmi.edu/SCRoftware/simplot/) [32] and the RDP4 software package [33]. Default parameters for the methods RDP [34], GENECOV [35], Bootscan [36], Chimaera [37], MaxChi [38], SiScan [39], and 3Seq [40] were used. Recombination events involving EV-D11-KAT-17-1263 and/or EV-D94-14-534 strains consistently identified by at least five of these seven RDP algorithms were further analyzed. The *p*-value cutoff was chosen as 0.05 and the best signals for recombination are associated with the lowest *p*-values, which indicates the approximate likelihood for the occurrence of exchange of sequences between genomes (recombination) rather than the probability of convergent evolution of the sequences.

## 3. Results

### 3.1. Identification by NGS of EV Strains Present in Water Concentrates 

NGS data obtained from Pan-EV RT-PCR products amplified from viral RNAs that were extracted from water concentrates were processed and analyzed as described in Materials and Methods. A high proportion of the total filtered NGS reads from all samples mapped to EV reference sequences (range: 84.07–99.22%) which shows the high specificity of the primers used for EV sequences present in RT-PCR products. A total of 93 contig sequences, corresponding to 93 different EV strains were identified in the 8 water concentrates from Nigeria. In most cases, nucleotide consensus sequences for each EV strain were approximately 4000 nucleotides in length, between nucleotides 553–4459 (numbering as in PV1 Sabin AY184219 reference strain) of the EV genome, covering part of the 5′-end non-coding region, the whole-capsid coding region and part of the P2 non-structural coding region. Nucleotide consensus sequences are available from NCBI sequence database with GenBank numbers MW373870 to MW373962. The results are shown in Figure 1 and Figure 2.

Samples contained EV strains belonging to 45 different serotypes and all four A, B, C, and D species with between 2 and 21 different serotypes identified in each sample. Species B strains were the most abundant (*n* = 43), followed by species C (*n* = 28), A (*n* = 20), and D (*n* = 2). We detected EV strains from 10 of 21 (47.62%), 26 of 62 (41.93%), 8 of 23 (34.78%), and 1 of 5 (20.00%), known serotypes for species A, B, C, and D, respectively. Ten EV serotypes were detected in Nigeria for the first time: CV-A2, CVA5, CV-A12, CV-A14, EV-B74, EV-B81, EV-B85, EV-B86, EV-B87, and EV-B106, two of them, CV-A12 and EV-B86, were first described in Africa. E-15 strains were the most widespread, present in five samples, whereas serotypes EV-A119, CV-A13, and CV-B4 were found in four samples each. E-15, EV-A119, and CV-A13 strains were found in all three locations. Some environmental EV strains identified corresponded to uncommon serotypes defined as those rarely associated with clinical cases and/or hardly ever reported elsewhere in the world from which very few nucleotide sequences are available, EV-A119, EV-A120, E-26, EV-B81, EV-B86, EV-B87, EV-C116, and EV-D111. Uncommon serotypes included A119 and D111 mostly seen in Central and South Africa. Several negative control samples were sequenced including RNA extraction, RT-PCR reaction, and water controls. No EV contig sequences were identified in any of the negative control samples following our analysis pipeline.

### 3.2. Nucleotide Sequence Analysis of Nigerian EV Environmental Strains

Appendix A shows the sequence comparison of EV strains from Nigeria with their closest relatives from the NCBI sequence database. In most cases, the closest relative was from Central-West Africa including Nigeria (*n* = 19) and neighboring/nearby countries such as Republic of Niger, Chad, Cameroon, Ivory Coast, Ghana, Senegal or DRC (*n* = 53). Although little sequencing data are available from recent EV clinical strains from Nigeria, the highest sequence similarity of the Nigerian strains from water samples was found to be with Nigerian clinical strains with few exceptions. In several cases, the observed VP1 sequence drift was compatible with EV strains of various serotypes having circulated in the country for long periods of time. We found sequence similarity between the EV strains from Nigeria and non-African EV strains; a CV-B4 strain from USA (96.0–96.4% VP1 sequence identity), an EV-A71 strain from Denmark (97.4% VP1 sequence identity), and a CV-A5 strain from France (96.6% VP1 sequence identity). Interestingly, the EV strain genetically closest to any of the Nigerian EV environmental strains (98.7% VP1 sequence identity) was found to be an EV-A76 isolate from a healthy chimpanzee found in 2016 in Nigeria. The closest E-15 strains from the water samples were also found to be distantly related to strains from non-human primates, gorillas and chimpanzees, found in Cameron in 2014. Nigerian EV strains from serotypes EV-A119 and EV-D111 were also found to be distantly related to viral sequences from non-human primates. 

Twelve different CV-A13 strains were identified in total; sample KAT-17-1263 contained 6 different CV-A13 strains and sample ADA-18-059 contained 3. Six different EV-A119 strains, 6 E-15, 4 E-14 and 4 CV-B4 strains were found in total. Phylogenetic trees based on VP1 nucleotide sequence comparisons of species A, B, C, and D EV strains are shown in Figure 3. We found VP1 sequence heterogeneity between EV strains of the same serotype in different samples indicating the presence of highly divergent cocirculating genetic lineages of several serotypes such as E-26, CV-A13, CV-A19, CV-A20, and CV-A24. Particularly, CV-A13 was found to be the most divergent serotype with several CV-A13 strains belonging to four different genetic clusters showing high sequence divergence between them (between 69.65–86.53% and 84.11–97.35% nucleotide and amino acid sequence identity in the VP1 region, respectively). On the contrary, CV-B4 strains found in the four samples from the Niger State collection site only, were very similar between them (98.10–99.88% VP1 nucleotide sequence identity).

As there is little sequence information available on EV-A119 genomes, we further analyzed the whole-capsid amino acid sequences of six EV serotype A119 strains, identifying two different genetic variants showing differences at several capsid amino acids some of which possibly represent variations at antigenic sites (Appendix A). In addition to whole-capsid sequences, we were able to explore the presence of a short alternative open reading frame (uORF) in the 5′-end region of EV strains’ genomes. This uORF, additional to the main ORF coding for all viral proteins, has recently been identified in some EVs [3], starts inside domain VI of the internal ribosome entry site and partly overlaps the VP4 coding sequence. Interestingly, we identified putative uORF amino acid sequences in 52 of the 93 EV strains belonging to species A (*n* = 6), B (*n* = 39) and C (*n* = 7) (Appendix A). As species D EVs remain poorly studied, we expanded our analysis beyond the capsid region by determining the whole genome sequence of an EV-D111 strain present in sample KAT-17-1263. In addition, with an aim to identify the possible recombinant events between species D EV serotypes and given that no other EV-D strains, apart from EV-D111, had recently been described in Nigeria, we determined the whole-genome sequence of an EV-D94 strain (14-534) available from a different project. This EV-D94 strain had been isolated from a paralytic case in the Republic of Niger (different to Niger State in Nigeria) in 2014. Katsina, where the EV-D111 KAT-17-1263 strain analyzed originates, is a northern State in Nigeria which shares ample border territory with the Republic of Niger. A previous study has shown clear evidence of different EV serotypes frequently crossing this border because of active people’s movements, with genetically close EV strains present in both countries [41], hence our choice to further characterize the EV-D94 14-534 strain. The results of these analyses are shown in Section 3.3, Section 3.4 and Section 3.5 below.

### 3.3. Whole Genome Determination of EV-D111 in Sample KAT-17-1263

The nearly complete genomic sequence of EV-D111, from nt 33 to nt 7353 (based on numbering of EV-D111 strain with GenBank number MT081371) was obtained by NGS analysis of overlapping RT-PCR products as described in Materials and Methods. The NGS analysis produced a main contig generating a consensus sequence of 7315 nucleotides in length. This contig contained 119,853 sequence reads producing high sequence coverage throughout the genome (mean coverage of 3028 reads per site) as shown in Figure 4 and very high sequence homogeneity as judged by single nucleotide polymorphism analysis [42] (data not shown). No uORF was identified in the complete EV-D111 genome. The sequence variation between the EV-D111 strain from Nigeria and previously sequenced EV-D111 isolates ranged between 86.7–91.5% and 97.1–98.1% at nucleotide and amino acid level, respectively. However, the sequence of the EV-D111 strain from water sample KAT-17-1263 was more closely related to an EV-D111 strain isolated from an AFP case in Nigeria in 2017 (GenBank number MH607128) for which a partial 5208 nt sequence is available, showing 96.6% and 98.8% sequence identity at nucleotide and amino acid level, respectively. The whole-genome sequence of EV-D111 KAT-17-1263 strain from Nigeria is available from NCBI sequence database with GenBank number MW384881.

### 3.4. Whole Genome Determination EV-D94 14-534 Strain from the Republic of Niger

The nearly complete genomic sequence of EV-D94 strain 14-534, from nt 1 to nt 7335 (based on numbering of strain with GenBank number MT081370) was obtained by NGS analysis of randomly synthesized PCR products as described in Materials and Methods. The NGS analysis produced a main contig generating a consensus sequence of 7293 nucleotides in length. This contig contained 44,105 reads with high sequence coverage throughout the genome (mean coverage of 1091 reads per site) as shown in Figure 4 and very high sequence homogeneity as judged by single nucleotide polymorphism analysis [43] (data not shown). No uORF was identified in the complete EV-D94 genome. The sequence variation between the 14-534 EV-D94 strain and previously sequenced EV-D94 isolates ranged between 85.3–88.9% and 97.6–98.0% at nucleotide and amino acid level, respectively, with the closest strain being an EV-94 strain from Egypt (GenBank number DQ916376). The whole-genome sequence of EV-D94 15-534 strain from the Republic of Niger is available from NCBI sequence database with GenBank number MW384880.

### 3.5. Recombination Analysis

The whole-genome sequences of EV-D111-KAT-17-1263 and EV-D94-14-534 strains were aligned to those from representatives of Species D EV serotypes to investigate the occurrence of recombination events among them. A main recombination event supported by high statistical significance values using six of the seven RDP algorithms was identified (average *p*-values of 3.52 × 10^−47^, 8.84 × 10^−55^, 1.05 × 10^−32^, 8.24 × 10^−15^, 1.08 × 10^−30^, and 6.35 × 10^−14^ for methods RDP, Bootscan, Maxchi, Chimaera, SiSscan, PhylPro, LARD, and 3Seq, respectively). EV-D94-14-534 was identified as a recombinant strain with parental sequences EV-D94 in the capsid region and EV-D111-KAT-17-1263 in the non-structural region and breakpoint identified at nucleotide 3802 of the sequence alignment. Figure 5 shows similarity plot and bootscanning analysis of EV-D94-14-534 strain conducted against representative strains of all known EV-D serotypes. 

This analysis confirmed that EV-D94-14-534 showed closer sequence similarity in the non-structural coding region to EV-D111 strains than to EV strains from other EV-D serotypes including EV-D94. These results strongly suggest that recombination events likely occur between EV-94 and EV-D111 serotypes. Both EV-D111-KAT-17-1263 and EV-D94-14-534 strains showed higher sequence similarity between themselves in the non-structural coding region than they did to EV strains from their respective serotypes while they showed closer sequence similarity to homotypic EV strains in the capsid coding region as expected. However, only a few whole-genome sequences from all EV-D serotypes are available so further research will be needed to better determine and understand these potential recombination events.

## 4. Discussion

High prevalence and diversity of human EVs were found in water samples from Nigeria. A total of 93 different EV strains from 45 different serotypes belonging to all four A, B, C, and D EV species were identified in eight water samples collected from water streams presumably contaminated by drainage channels situated in three locations in Nigeria, confirming the high risk associated with domestic use of such water supplies due to sewage contamination. We were able to identify ten EV serotypes detected in Nigeria for the first time, two of them, CV-A12 and EV-B86, firstly described in Africa. EV strains were identified from both serotypes commonly associated with clinical syndromes and those that appear to circulate silently but could eventually cause outbreaks and disease. Among them, several emerging serotypes, rarely reported elsewhere, were detected such as EV-A119, EV-B87, EV-C116, and EV-D111. E-15 strains were the most prevalent, present in five samples, whereas serotypes EV-A119, CV-A13, and CV-B4 were found in four samples each. 

Our data revealed close phylogenetic relationships of the Nigerian strains with EV strains reported elsewhere, mostly from Central West Africa, indicating that many of these serotypes are endemically circulating in the region. High sequence divergence from the closest relative in some cases suggests long-term silent transmission of specific EV serotypes and possible gaps in surveillance. We found sequence similarity between the EV strains from Nigeria and non-African EVs from serotypes CV-B4, EV-A71, and CV-A5 which could represent recent importation events. EV-A71, in particular, has been associated with neurological disease [7]. The EV-A71 strain found in sample KAT-17-1263 belongs to genogroup C1 known to circulate worldwide and different to genogroups E and F endemic in Africa [44]. Our data showed the presence of highly divergent cocirculating genetic lineages of several serotypes, particularly CV-A13, for which strains belonging to four different genetic clusters were identified showing high sequence divergence between them. This indicates widespread presence of this serotype in the country probably dating back many years as it has been shown in other African countries before [45]. Divergent cocirculating strains were also observed for serotypes CV-A19, CV-A20, and CV-A24 all belonging to EV species C like CV-A13 with the ability to recombine with poliovirus likely contributing to the generation and evolution of vaccine-derived poliovirus strains that have been shown to cause polio outbreaks particularly in Africa [46,47]. On the contrary, CV-B4 strains found in one location only, were very similar between them, suggesting sustained local circulation. Some of the EV strains were found to be genetically related to EV isolates from non-human primates providing further evidence for cross-species transmission of some EV serotypes as extensively documented before [48].

Our study further illustrates the power of using next generation sequencing analysis allowing the identification of multiple EV strains in a single sample [14,15,16,17,18,19] and overcoming the limitations of traditional surveillance methods based on cell culture virus isolation and Sanger sequencing [49]. The number of serotypes identified in this study far exceeds those found in previous studies. As a comparison, only 9 EV strains from 5 different serotypes and 25 EV strains from 7 different serotypes were identified in previous studies involving 9 and 8 comparable water samples collected in 2010 [50] and 2012-2013 [51] in Nigeria, respectively. As shown in Appendix A, a total of 71 different EV serotypes have been identified in Nigeria in recent years with 6 serotypes only identified in stool samples from AFP cases (CV-A9, CV-B1, E-27, E-30, EV-B111, and EV-D94), 5 in stool samples from healthy individuals (CV-A3, CV-A8, E-2, E-32, and EV-B88) and 10 in environmental samples as referred above. 

Although any amplification-based method such as the whole-capsid RT-PCR assay described here may contribute to bias toward specific strains or serotypes and may risk missing some strains due to mismatches in primer binding sequence regions, we have so far been able to sequence in different studies EV strains from 102 of the 110 different EV serotypes that have been described [16,17,19]. The use of environmental surveillance also contributes to expanding the diversity of EV serotypes that can be detected, improving our understanding of EVs circulating in human populations, as most EV infections appear to be asymptomatic or their detection requires timely and adequate sample collection and analysis. In addition, being able to obtain whole-capsid nucleotide sequences of circulating EV strains allows a more in-depth sequence analysis to establish more accurate temporal and geographical associations between environmental and clinical EV strains from the same or different locations. Furthermore, whole-capsid amino acid sequences provide all necessary information that make up viral particles which contain determinants of critical viral properties such as antigenic structure, receptor-binding, and virulence. This information is essential to better understand the biology of EVs associated with human disease, identifying virus evolution patterns, and contributing to the design of antivirals and vaccines. We also examined the presence of an alternative short uORF, recently described for some EVs [3] which is located upstream the main ORF coding for all viral proteins and identified such region in 52 of the Nigerian EV water strains found in the Nigerian water. The product of this uORF was shown not to be critical for the replication of E-7 but improved the growth of this virus in gut epithelial cells suggesting a possible role in cell tropism [3].

Nucleotide sequence information at the 3′-end of the capsid coding region can also be used to design EV strain-specific internal primers to synthesize overlapping PCR products and obtain whole-genome sequences of selected EV strains as we have shown here. Analysis of the whole-genome sequences of an EV-D111 strain from a water sample in Nigeria and an EV-D94 strain isolated from a paralytic case in the Republic of Niger as well as representative sequences of all known EV-D serotypes revealed the occurrence of recombination events among species D EVs, a common feature of species A, B, and C EVs. This is not unexpected as frequent EV transmissions are known to occur across the border between the Republic of Niger and Nigeria [41], and especially in border states like Katsina where the EV-D111 strain was detected. Evidence of evolution of EV-D94 and EV-D111 through intertypic recombination has been shown before in a recent study [20]. Our analysis identified the recombination breakpoint to be located in the first half of the P2 region, known as a frequent recombination hotspot in species A, B, and C EV recombinant strains [44,52,53,54].

## 5. Conclusions

We found high prevalence and diversity of human EV serotypes in water samples from Nigeria highlighting the high risk of infection associated with domestic use of contaminated water supplies. Our results show that setting up ES systems in combination with high resolution molecular analysis will help determining the incidence of EV transmission in different populations, being able to monitor the resolution of outbreaks, evaluate the impact of vaccines on virus circulation, and establish alert systems for the early detection of specific EV serotypes known to be associated with human disease.

## Figures and Tables

**Figure 1 viruses-13-00249-f001:**
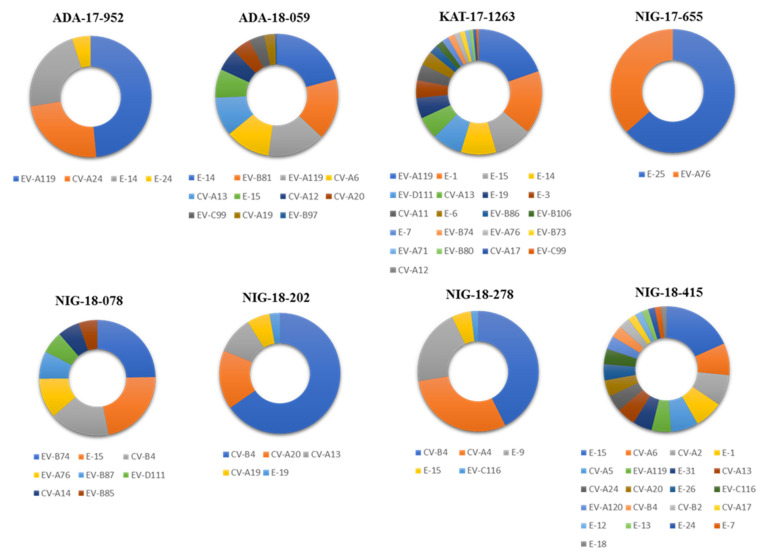
Enteroviruses (EV) serotypes identified in water samples from Nigeria. Filtered next generation sequencing (NGS) reads from RT-PCR products from the water concentrates were mapped to consensus sequences obtained for each sample. The proportions of reads mapping to each EV serotype in each sample are shown with different colors as indicated at the bottom of the figure.

**Figure 2 viruses-13-00249-f002:**
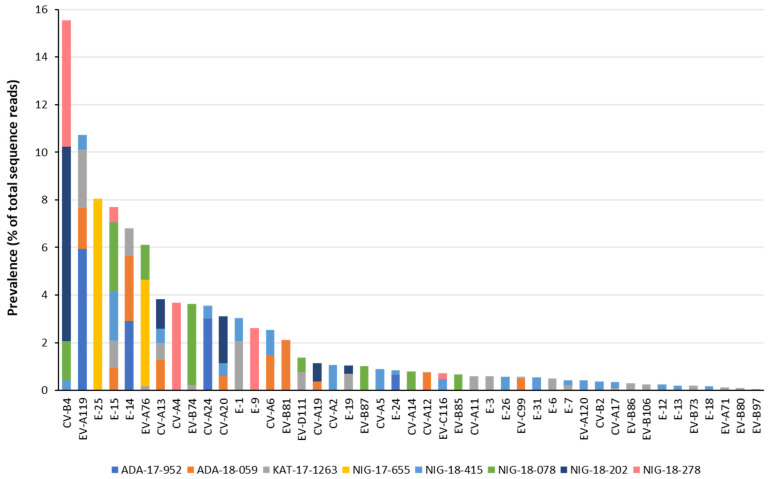
Overall prevalence of EV serotypes in water samples from Nigeria. Filtered NGS reads from RT-PCR products from the water concentrates were mapped to consensus sequences obtained for each sample. The proportions of NGS reads mapping to each EV serotype are shown. The sample source is shown in different colors as indicated at the bottom of the figure.

**Figure 3 viruses-13-00249-f003:**
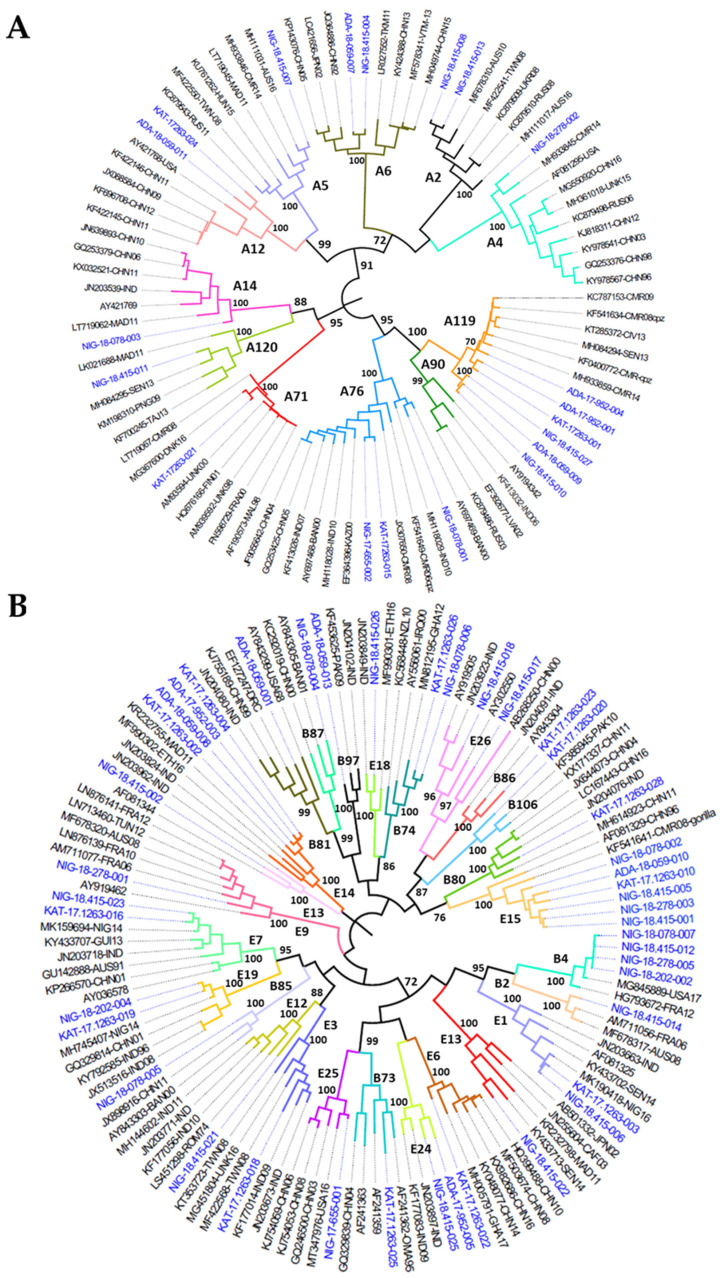
Phylogenetic analysis of VP1 sequences of EV strains from species (**A**–**D**) (panels **A**–**D** in the figure) identified in water samples from Nigeria. Phylogenetic relationships between EV sequences were inferred by using the Maximum Likelihood method and Tamura-Nei model using MEGA X software [29]. Initial tree(s) for the heuristic search were obtained automatically by applying Neighbor-Join and BioNJ algorithms to a matrix of pairwise distances estimated using the Maximum Composite Likelihood (MCL) approach, and then selecting the topology with superior log likelihood value. The tree is drawn to scale, with branch lengths measured in the number of substitutions per site. The percentage of replicate trees in which the associated taxa clustered together in the bootstrap test (1000 replicates) are shown next to the branches. VP1 sequences from Nigerian strains in this study are shown in blue text. Abbreviations for country names are shown in the Abbreviations section at the bottom of the manuscript.

**Figure 4 viruses-13-00249-f004:**
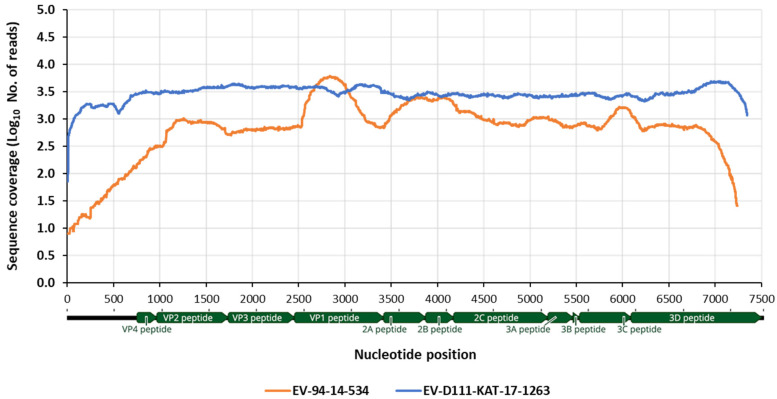
Genome coverage of EV-D94-14.534 strain from the Republic of Niger (orange line) and EV-D111-KAT-1263 strain from Nigeria (blue line) by NGS analysis. Filtered reads were mapped to the final consensus sequences generated by de novo assembly. The number of sequence reads at each nucleotide position is shown. The location of viral genes is indicated.

**Figure 5 viruses-13-00249-f005:**
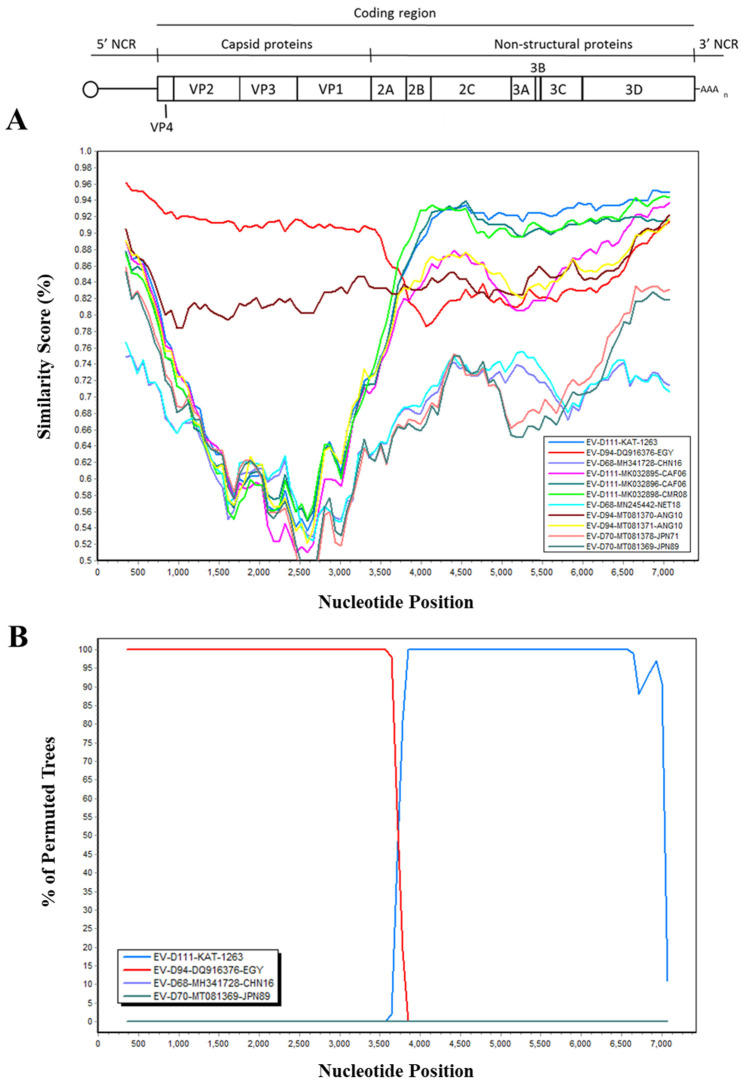
Recombination analysis of EV-D94-14-534 strain from Niger. Plot of similarity (**A**) and bootscanning analysis (**B**) using EV D whole-genome sequences from the NCBI sequence database. The EV genetic map is shown in the top panel. The analyses were conducted using SimPlot version 3.5.1 with the Kimura distance method and a sliding window of 700 base pairs moving in steps of 70 nucleotides. The genome of EV-D94-14-534 strain served as a query sequence.

**Table 1 viruses-13-00249-t001:** Details of water samples analyzed.

Sample No.	Location ^1^	Collection Date
ADA-17-952	Adamawa	August 2017
ADA-18-059	Adamawa	January 2018
KAT-17-1263	Katsina	October 2017
NIG-17-655	Niger	June 2017
NIG-18-078	Niger	January 2018
NIG-18-202	Niger	February 2018
NIG-18-278	Niger	March 2018
NIG-18-415	Niger	April 2018

^1^ It must be clarified that the Niger location corresponds to a Nigerian State, different to the country Republic of Niger.

## Data Availability

Nucleotide sequences determined in this study are available from NCBI sequence database with GenBank numbers MW373870 to MW373962, MW384880 and MW384881.

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
