# Peer review of "High Diversity of Human Non-Polio Enterovirus Serotypes Identified in Contaminated Water in Nigeria"

_viruses, 2021, doi:10.3390/v13020249_

Round 1

Reviewer 1 Report

Extremely interesting and useful article, high methodological level.

Technical remarks only.

1) line 220: apparently extracted;

2) line 298: omitted CV-B4 strains;

3) line 323: apparently 3.3-3.5

Author Response

Comments from reviewer

Extremely interesting and useful article, high methodological level.

Response from authors

The authors thank the reviewer for the positive assessment and helpful remarks. Responses to comments and corrections of manuscript are described below.

Comments from reviewer

Technical remarks only.

  • line 220: apparently extracted;
  • line 298: omitted CV-B4 strains;
  • line 323: apparently 3.3-3.5

The text has been corrected as suggested:

  • The word “extracted” was corrected (Line 217 in the “Nigeria PanEV Viruses v2 tracked.pdf” file)
  • The term “CV-B4” was corrected (Line 528 in the “Nigeria PanEV Viruses v2 tracked.pdf” file)
  • “3.3-3.5” was corrected (Line 553 in the “Nigeria PanEV Viruses v2 tracked.pdf” file)

Reviewer 2 Report

In the submitted paper, Majumdar et al. reported the molecular detection and epidemiology of EVs collected from sewage samples in Nigeria. The manuscript is straightforward and provide genetic information of the circulating virus in Africa where such studies are a very few. Still, I have several concerns in the study for the publication.

Major points

  1. Recombination analysis was performed assuming that EV-D111 KAT-17-1263 is a recombinant. Although Figure 5A shows fluctuation of similarity between that strain and EV-D94 viruses, the similarity is constantly high with other EV-D111 viruses. That suggests recombination occurred in not EV-D111 KAT-17-1263 but other viral strains used as references in Figures 5A and 5B. The authors must test other tools to detect recombination that can analyze not only where (genomic position) recombination occurs but also which strain is recombinant. Examples include RDP4. Please first show the evidence that EV-D111 KAT-17-1263 is recombinant.

  1. Please discuss the similarity and difference in prevalence of EV serotypes in the country observed in clinical samples (from national surveillance data?) and those found in sewage shown in the present study.

Minor points

  1. The authors used “direct sequencing” and “direct detection” for their methodology in the Abstract, Introduction, and Discussion. They used the words to indicate their methodology did not rely on cell culture. However, in molecular testing, “direct sequencing” means analysis of nucleic acid without amplification step (i.e., PCR). That is not the case for the present study. The authors should not use “direct” to avoid the confusion.

  1. The Abstract needs background information of the study.

  1. (Line 29) Although the authors mentioned the result of whole-genome analysis, they did not describe they had conducted that analysis in the Abstract before.

  1. At the end of Introduction, the authors had better briefly describe the objects of the present study and what they have conducted.

  1. (Line 104) NIBSC should be spelled out.

  1. (Line 171 and Figure 3) If the authors are interested in evolutionary “history”, phylogenetic trees must be constructed using ML method. Neighbor-joining is good for grouping but not for inferring evolutionary path.

  1. (Line 323) “2.3-2.5” should be “3.3-3.5”.

Author Response

Comment from reviewer
In the submitted paper, Majumdar et al. reported the molecular detection and epidemiology of EVs collected from sewage samples in Nigeria. The manuscript is straightforward and provide genetic information of the circulating virus in Africa where such studies are a very few. Still, I have several concerns in the study for the publication.

Response from authors
The authors thank the reviewer for the positive assessment and helpful remarks.Responses to comments and corrections of manuscript are described below.

Comment from reviewer
Major points
1.Recombination analysis was performed assuming that EV-D111 KAT-17-1263 is a recombinant. Although Figure 5A shows fluctuation of similarity between that strain and EV-D94 viruses, the similarity is constantly high with other EV-D111 viruses. That suggests recombination occurred in not EV-D111 KAT-17-1263 but other viral strains used as references in Figures 5A and 5B. The authors must test other tools to detect recombination that can analyze not only where (genomic position) recombination occurs but also which strain is recombinant. Examples include RDP4. Please first show the evidence that EV-D111 KAT-17-1263 is recombinant.

Response from authors
Given the limited number of sequences available in public databases, analysis of recombination between enteroviruses can only give an approximation on the subject. Closely related parental strains can be rarely identified but finding evidence for a possible recombination structure is possible. We conducted a more thorough recombination analysis using the software package RD4 as suggested by the reviewer. The results indicate that it is more likely that the actual recombinant virus is EV-94-15-534 that we sequenced from an AFP sample rather than EV-D111-KAT-1263 found in a water sample from Nigeria. However, the overall message remains the same in that exchange of genetic material occurs among species D EV serotypes with EV-94-15-534 strain showing close genetic similarity to other EV-D94 strains as expected but closer sequence similarity to EV-D111-KAT-1263 in the non-structural genomic region. The manuscript has been modified to reflect this additional analysis in section 2.8 (Materials and Methods) and section 3.5 (Results) as well as a slight modification of the text in the Discussion section (lines 878-881 in the “Nigeria PanEV Viruses v2 tracked.pdf” file). We than the reviewer for this helpful suggestion that has improved the quality of the manuscript.

Comment from reviewer
2.Please discuss the similarity and difference in prevalence of EV serotypes in the country observed in clinical samples (from national surveillance data?) and those found in sewage shown in the present study.

Response from authors
We have added a supplementary Table S2 showing the distribution of EV serotypes identified in different surveillance studies in Nigeria and discussed the results in the Discussion section as suggested (lines 844-852 in the “Nigeria PanEV Viruses v2 tracked.pdf” file). As a consequence of this review, we have slightly modified the number of new serotypes found in this study from thirteen to ten.

Comment from reviewer
3.The authors used “direct sequencing” and “direct detection” for their methodology in the Abstract, Introduction, and Discussion. They used the words to indicate their methodology did not rely on cell culture. However, in molecular testing, “direct sequencing” means analysis of nucleic acid without amplification step (i.e., PCR). That is not the case for the present study. The authors should not use “direct” to avoid the confusion.

We have corrected the text throughout the manuscript as suggested to avoid confusion.

Comment from reviewer
4.The Abstract needs background information of the study.

Response from authors
We have added a short background of the study at the beginning of the Abstract section (lines 21-25 in the “Nigeria PanEV Viruses v2 tracked.pdf” file).

Comment from reviewer
5.(Line 29) Although the authors mentioned the result of whole-genome analysis, they did not describe they had conducted that analysis in the Abstract before.

Response from authors
This is now mentioned in the abstract (lines 33-34 in the “Nigeria PanEV Viruses v2 tracked.pdf” file).

Comment from reviewer
6.At the end of Introduction, the authors had better briefly describe the objects of the present study and what they have conducted.

Response from authors
A short description of the objectives of the study and work done has been added at the end of the Introduction section (lines 118-150 in the “Nigeria PanEV Viruses v2 tracked.pdf” file).

Comment from reviewer
7.(Line 104) NIBSC should be spelled out.

Response from authors
NIBSC has been spelled out in the author’s list and the main text as suggested (lines 204-205 in the “Nigeria PanEV Viruses v2 tracked.pdf” file).

Comment from reviewer
8.(Line 171 and Figure 3) If the authors are interested in evolutionary “history”, phylogenetic trees must be constructed using ML method. Neighbor-joining is good for grouping but not for inferring evolutionary path.

Response from authors
We agree that using phylogenetic grouping or phylogenetic relationships reflects better the analysis that we conducted and what we intended to show. The text in section 2.5 (Materials and Methods) and Figure 3 legend have been modified accordingly.

Comment from reviewer
9.(Line 323) “2.3-2.5” should be “3.3-3.5”.

Response from authors
The text has been modified as suggested (line 553 in the “Nigeria PanEV Viruses v2 tracked.pdf” file).

Reviewer 3 Report

In this paper the authors describe a large set of human enterovirus (EV) strains belonging to 45 different serotypes in drainage channels in Nigeria. 

Environmental surveillance is important as it provide additional information on the circuskation of specific non-polio strains within sewage water.

Minor edits:

Introduction: as the paper is not focusing on associations between symptomatology and specific genotypes, I would advise to delete the following paargrpah in the introduction:

Being able to correctly associate disease syndromes with EV infection, the need to collect adequate and timely samples from patients and the use of sensitive and specific methods for EV detection and identification are also critical factors that may have limited the number of EV strains that have been reported to date [9].

Methods

The authors state:

Samples that did not contain poliovirus were specifically selected in order to focus on non-polio EV serotypes and avoid biosafety containment issues. I was wondering how many of the sewage water samples container poliovirus. Was there and difference in the collection date or processing methods of sewage water samples containing plooivirus versus the water samples not containing poliovirus? Could there be a bias?

Overall nice work

Author Response

Comment from reviewer
In this paper the authors describe a large set of human enterovirus (EV) strains belonging to 45 different serotypes in drainage channels in Nigeria. Environmental surveillance is important as it provide additional information on the circulation of specific non-polio strains within sewage water.

Response from authors
The authors thank the reviewer for the positive assessment and helpful remarks. Responses to comments and corrections of manuscript are described below.

Comment from reviewer
Minor edits: Introduction: as the paper is not focusing on associations between symptomatology and specific genotypes, I would advise to delete the following paragraph in the introduction: Being able to correctly associate disease syndromes with EV infection, the need to collect adequate and timely samples from patients and the use of sensitive and specific methods for EV detection and identification are also critical factors that may have limited the number of EV strains that have been reported to date [9].

Response from authors
The paragraph has been removed from the Introduction section as suggested.

Comment from reviewer
Methods
The authors state: Samples that did not contain poliovirus were specifically selected in order to focus on non-polio EV serotypes and avoid biosafety containment issues. I was wondering how many of the sewage water samples container poliovirus. Was there any difference in the collection date or processing methods of sewage water samples containing poliovirus versus the water samples not containing poliovirus? Could there be a bias?

Response from authors
This is an interesting question that might need further research to be fully responded. However, we think we have sufficient technical and scientific evidence to say that it is unlikely that there is a bias in the distribution of non-polio EV serotypes in water samples analysed in this study due to the selection of poliovirus-negative samples for our analysis. First, the workflow used for all samples was identical, as described in the WHO Guidelines for environmental surveillance of poliovirus circulation. 2003, (WHO/V&B/03.03). Secondly, both polio and non-polio enteroviruses have similar structural and physicochemical properties and are expected to be affected by dilution and inactivation effects in sewage in a similar manner. Absence of any enterovirus detection is used and has proven to be a good indicative of poor environmental sensitivity for poliovirus detection regardless of EV serotypes present in the sample. Lastly, recent studies suggest there is little interference between the circulation of different EV serotypes in the same population with circulation of specific EV serotypes not having a major impact on the transmission of other EV serotypes. Indeed, several other factors do seem to have an effect on the detection of EVs in environmental samples such as the nature of the sewage network, the appropriateness of the sampling site, the catchment population and the physicochemical properties of sewage. However, it is unlikely that these properties would affect different EV serotypes differently for the reasons stated above, although it cannot completely be ruled out. Current WHO guidelines recommend establishment of environmental surveillance sites where there is a convergent sewage network and a catchment population of 100 000 to 300 000 people. However, most areas at high risk of poliovirus transmission have informal drainage and sewerage and sampling sites, such as the ones used in this study, might be very different between them making comparisons between sites more difficult. Samples used in this study were taken from water streams used by some communities that are often contaminated with sewage, but levels of contamination and catchment population numbers are difficult to estimate. We have added a paragraph in section 2.1 (Materials and Methods) to state why we don’t expect bias in EV serotype distribution due to selecting poliovirus-negative samples.

Round 2

Reviewer 2 Report

My concerns raised in the first round of review are well addressed. I appreciate the authors' hard work.